# Evaluation of Qualitative Dietary Protocol (Diet4Hashi) Application in Dietary Counseling in Hashimoto Thyroiditis: Study Protocol of a Randomized Controlled Trial

**DOI:** 10.3390/ijerph16234841

**Published:** 2019-12-02

**Authors:** Natalia Wojtas, Lidia Wadolowska, Elżbieta Bandurska-Stankiewicz

**Affiliations:** 1Department of Human Nutrition, University of Warmia and Mazury in Olsztyn, Sloneczna 45F, 10-718 Olsztyn, Poland; lidia.wadolowska@uwm.edu.pl; 2Department of Internal Medicine, School of Medicine, University of Warmia and Mazury, 10-718 Olsztyn, Poland; bandurska.endo@gmail.com

**Keywords:** adiposity, dietary intervention, dietary protocol, dietetic counseling, diet quality, Hashimoto, hypothyroidism, metabolic parameters, obesity, quality of life

## Abstract

The current state of knowledge related to diet in Hashimoto thyroiditis (HT) is far from satisfactory, as many HT subjects experience several disorders and report reduced quality of life. There are three aims of the study: (1) to develop a qualitative dietary protocol (QDP; ‘Diet4Hashi’) as a simple, graphic–text tool dedicated to TH subjects, (2) to evaluate the use of the QDP in dietetic counseling compared to conventional dietetic counseling (CDC) in HT women, and (3) to assess the impact of both the QDP and the CDC on the diet quality, quality of life, adiposity, and metabolic parameters of HT women. The QDP is based on subject self-monitoring supported with a graphic–text tool to help them in food selection and adequate food frequency consumption, while the CDC on oral explanation and printed sample menus were provided by a dietician. The QDP contains two lists: (A) foods recommended for consumption and (B) foods with limited consumption, along with indicated consumption frequency per day/week/month. Both approaches include the same dietary recommendations for HT extracted from the literature but differ in subject–dietician cooperation. To summarize the evidence regarding dietary recommendations in HT, the PubMed, Embase, and Cochrane Library databases (to March 2019) and the bibliographies of key articles were searched. The study is designed as a dietary intervention lasting six months in two parallel groups: experimental and control. In the experimental group, the QDP will be applied, while in the control group, the CDC will be applied. In total, the study will include a baseline of 100 women with diagnosed HT. The subjects will be randomly allocated into the experimental/control groups (50/50). Data related to diet quality and other lifestyle factors, nutrition knowledge, quality of life, thyroid function, body composition, blood pressure, serum fasting glucose, and lipid profile at baseline and after a six-month follow-up will be collected. This study was conducted to develop a dietary protocol (Diet4Hashi) that is easy to follow for HT subjects, and it will contribute to providing valuable data that are useful to dieticians and physicians. It is anticipated that this graphic–text qualitative dietary protocol, by improving food selection and diet quality, may reduce adiposity and improve metabolic parameters and the quality of life of HT women.

## 1. Introduction

Chronic lymphocytic thyroiditis (Hashimoto thyroiditis, Hashimoto, HT) is the most frequent autoimmune disorder and the cause of hypothyroidism in iodine-sufficient countries [1,2,3]. The prevalence of primary hypothyroidism affects 1%–3% of the population, while subclinical hypothyroidism is more common (estimated at 4%–10%). Abnormal functioning of the thyroid gland is diagnosed in 22% of the population [4,5,6]. In Poland, almost 1.5 million cases of thyroid gland diseases were diagnosed in 2004–2009 in surveys on women’s health [7,8]. Women had thyroid gland disease almost nine times more often than men. Within five years, the number of women declaring thyroid disease increased by 1.4 percentage points in 2009 compared to 2004. Data on HT prevalence in Poland is missing. Findings from the National Health and Nutrition Examination Survey (NHANES) III showed that the prevalence of HT was 4.8 per 1000 USA residents [9]. 

Thyroid hormones control growth, metabolism, and body development and take part in the production of structural proteins, enzymes, and other hormones. The majority of subjects with thyroid disease experience problems in maintaining normal body weight and have a higher body mass index and waist circumference than healthy subjects [10]. Herwig et al. [11] showed that the total metabolic rate can be lowered by up to 50% in serious cases of hypothyroidism. Lowering the total metabolic rate supports body weight gain and predisposes patients to obesity, even when a patient’s physical activity and dietary energy load are the same as before hypothyroidism diagnosis. Incorrect energy conversion worsens the functioning of most body tissues and disturbs metabolism, including glucose metabolism [12]. Findings from the Polish study (2001–2010) showed that HT co-occurred with diagnosed diabetes in over 27% of HT subjects and fasting blood glucose level or impaired glucose tolerance in about 17% of HT subjects [13]. Furthermore, even after normalizing thyroid function via hormonal replacement, many HT individuals persist with numerous symptoms, such as chronic fatigue and irritability, dry skin, hair loss, nervousness, and impaired quality of life [14].

Medical management with thyroid hormone replacement is the fundamental treatment in HT [15,16]. Nutrition can support the treatment and the impact of diet on thyroid function cannot be denied, as dietary micronutrients play a role in thyroid hormones synthesis. To date, there is limited information from well-evidenced studies, and none of the scientific societies have developed comprehensive dietary recommendations on HT [17,18,19,20,21]. The recommendations of the American Thyroid Association [19] and the American Dietetic Association [20] refer exclusively to thyroid cancer and the intake of iodine and salt. A recent review by Liontris [18] described the role of iodine, selenium, vitamin D, and gluten in the dietary management of HT patients. A gluten-free diet was considered potentially beneficial for HT patients independently of a comorbid diagnosis of celiac disease [17,18]. However, it was speculated that in subjects ‘on a diet’, their quality of life may be negatively impacted due to the restrictive nature of a gluten-free diet [17,18,21]. Abbott et al. [22] reported an improvement in health-related quality of life and clinical symptom burden in TH women after implementation of a phased elimination diet known as the Autoimmune Protocol supported by a multi-disciplinary team with a 10-week online diet and lifestyle coaching. Despite this, no significant changes in thyroid function or thyroid antibodies were found. Thus, many dieticians have difficulties in summarizing study findings and developing practical dietary recommendations. On the other hand, there has been growing interest in dietary counseling, due to an increasing number of HT subjects. Patients who are weakly supported by dieticians or other health professionals seek advice from other sources, e.g., mass media and the Internet, including those from less credible sources and/or giving contradicting information. 

### 1.1. Research Objective and Hypothesis

The current state of knowledge related to diet in HT is far from satisfactory, while many HT subjects experience many disorders and report reduced quality of life [13,14,18,23]. There is a strong need to summarize the evidence and develop dietary recommendations that are simple for HT subjects to understand and implement daily over a long-term period, and are useful for dieticians and other health professionals. The conventional approach to dietetic counseling is a structured process aimed at supporting individual patients to modify their dietary behaviors to improve health outcomes [24]. Conventional dietetic counseling is based on the development of individual recommendations and formulating detailed guidelines, which may prove difficult for patients to follow over the long term due to growing fatigue, lowering of motivation, the time required, and difficulties maintaining subject compliance to the diet [25]. Mitchell et al. [24] reported that dietetic counseling appears to be effective, e.g., in improving diet quality, diabetes outcomes, and weight loss outcomes, while limited evidence was found for the effectiveness of direct dietetic counseling alone in achieving outcomes relating to plasma lipid levels and blood pressure. The authors suggested that future research might explore novel nutrition counseling approaches.

There are three aims of the study: (1) to develop a qualitative dietary protocol (QDP; ‘Diet4Hashi’) as a simple, graphic–text tool dedicated TH subjects, (2) to evaluate the use of the QDP in dietetic counseling compared to conventional dietetic counseling (CDC) in HT women, (3) to assess the impact of both the QDP and the CDC on diet quality, quality of life, adiposity, and metabolic parameters. It is planned to develop a food-only protocol without considering the use of supplements because an adequate, well-balanced diet should be the first step in a dietary approach to improving health.

The research hypotheses are:Dietetic counseling with the use of a qualitative dietary protocol can facilitate patient understanding and long-term adherence to dietary recommendations compared to conventional dietetic counseling.A well-composed diet supported with a simple, qualitative dietary protocol, by improving food selection and diet quality, can improve the quality of life of HT subjects, reduce adiposity, and improve metabolic parameters. Furthermore, a better effect of a qualitative dietary protocol than conventional dietetic counseling is possible.

### 1.2. Study Design

The study is designed as a single-center dietary intervention lasting six months, with a random allocation of HT women into two parallel groups: experimental and control. In the experimental group, the QDP will be applied, while the CDC will be applied in the control group. The QDP is based on subject self-monitoring supported with a graphic–text tool to help them in food selection and adequate food frequency consumption, while the CDC will be supported by an oral explanation and printed sample menus given by dietician. Both approaches include the same dietary recommendations for HT, extracted from the literature, but differ in subject–dietician cooperation. 

The schedule of the enrolment, intervention, and data collection is presented in Table 1. All data will be collected at the same time for both groups, with one exception. The experimental group, supported with a graphic–text tool, will be asked to complete self-monitoring diaries during the six months of the study.

## 2. Materials and Methods

### 2.1. Ethical Approval

The study protocol was registered and approved by the Bioethics Committee of the Faculty of Medical Sciences, University of Warmia and Mazury in Olsztyn on 17 June 2010 and 25 April 2019, Resolution No. 20/2010 and Resolution No 43/2019). Written informed consent to participate was obtained.

### 2.2. Participants Selections

The participants will be recruited for the study through ads in the press, via the Internet and by e-mail invitation between October and December 2019. Approximately 140 women with HT are expected to respond to the invitation letter (describing the details of the planned trial) and will be screened for eligibility (Figure 1). All visits for data collection will take place at the Department of Human Nutrition (Center of Gastronomy with Dietetics and Food Bioassessment), Faculty of Food Science, University of Warmia and Mazury in Olsztyn. The trial starts in winter months, and all subjects will be advised to continue their long-term vitamin D supplementation. 

To take part in the study, the participants have to be prior diagnosed as having HT according to the American Thyroid Association guidelines [26] and have a medical certificate for levothyroxine or other thyroid hormone treatment (e.g., levothyroxine plus liothyronine). The participants will be required to show a medical diagnosis with thyroid ultrasonography and the results of thyroid function tests carried out no later than two months before enrolment. Detailed inclusion and exclusion criteria are presented in Table 2. Information collected at the interview, which not will be used as including/excluding criterion, is the age at which HT was diagnosed, the use of dietary supplements in the past, the dose of levothyroxine or other thyroid hormones in current treatment, and hormone status (menstruating: yes, no, irregular). Elevated serum auto-antibodies against thyroid peroxidase and thyroglobulin will be recorded but will not be used as excluding criterion.

### 2.3. Randomization and Blinding

Patients will be randomly allocated to either the experimental group (QDP; approximately n = 50) or the control group (CDC; approximately n = 50). Stratified randomization will be based on age and will be carried out by a researcher not directly involved in the study. The patients will be unaware of whether they belong to the experimental or the control group. The researchers and laboratory personnel who will be involved in data collection and measurements (excluding the researcher who will be involved in dietetic counseling) will not know if a participant is included in the QDP group or the CDC group. The study is designed as a single-blinded study without a placebo.

### 2.4. Dietary Intervention with QDP and CDC

Both dietetic approaches (QDP and CDC) are based on the same dietary recommendations in HT. To identify studies reporting dietary recommendations in HT or hypothyroidism (reviews) and investigations on diet and health outcomes in HT or hypothyroidism (original papers), the PubMed, Embase, and Cochrane Library databases (to March 2019) and the bibliographies of key articles were searched. Based on the data extracted, dietary recommendations for HT were established. Justification for these recommendations is presented in detail in the discussion (Section 3.1 and Section 3.2). 

General dietary guidelines for HT subjects will be distributed as a printed leaflet, which was the same for all participants in both the QDP and CDC groups (Appendix A, in Polish: Appendix A). In the leaflet, there are universal dietary guidelines which, if adhered to, may reduce the risk of health disorders. Similar pro-healthy dietary guidelines are widely promoted for the general population [27,28,29]. 

In the CDC group, dietetic counseling will be conducted in a conventional manner. Respondents will receive detailed oral explanations and printed sample menus given by a dietician. In this model of subject–dietician cooperation, the dietician is more involved while the subject is less (being more passive).

In the QDP group, dietetic counseling will be supported with a simple tool containing two lists: (A) foods recommended for consumption (seven items) and (B) foods with limited consumption (six items), both with an indicated consumption frequency per day/week/month (Table 3 and Table 4, in Polish: Appendix A). To develop both lists, we summarized data collected from the literature search and extracted recommended frequencies of food consumption, dividing foods on those recommended for more frequent consumption (list A) or less frequent consumption (list B). If the state of knowledge was unclear or findings were contradictory or poorly documented, then guidelines for the general population were adopted [30] to establish consumption frequency per day/week/month. Each food list is developed as a graphic–text self-monitoring diary to help patients in food selection and to exercise self-monitoring of food consumption and facilitate compliance with dietary guidelines. The printed version of the self-monitoring diary will be distributed, and participants will be asked to complete a diary for six months and return it during a visit. Respondents’ adherence to dietary guidelines will be expressed in points; one point per each compliance to each recommendation will be assigned. All points will be summarized to express an adherence score to the QDP an Adherence Score to the QDP (AdhS; ranged 0-13 points). Data from six months will be collected, and the mean AdhS will be calculated. The AdhS will be considered on three levels—low (0–4 points), average (5–9 points), and high (10–13 points)—and interpreted; more points indicates better compliance with dietary guidelines.

### 2.5. Diet Quality 

To collect dietary data, the semi-quantitative food frequency method will be applied, and a food frequency questionnaire (KomPAN) will be used. The KomPAN questionnaire consists of 46 food items and includes a comprehensive variety of foods usually consumed in Poland [30]. The KomPAN questionnaire has been assessed in healthy/unhealthy subjects as reliable tools with acceptable to very good reproducibility [30,31]. To assess diet quality, two predefined diet quality scores (the a priori approach) will be used: pro-Healthy-Diet-Index (pHDI) and non-Healthy-Diet-Index (nHDI).

Participants will be asked to indicate the average frequency of consumption of each food item over the 12 last months. Enrolled participants will receive a questionnaire from trained researchers, who will provide guidance and assistance as required on a one-to-one basis. 

To indicate food frequency consumption, respondents will choose one of six categories (converted into daily frequency): ‘never’ (0 times/day), ‘1–3 times a month’ (0.06 times/day), ‘once a week’ (0.14 times/day), ‘few times a week’ (0.5 times/day), ‘once a day’ (1.0 time/day), or ‘few times a day’ (2.0 times/day) [31]. Daily frequencies of 24 food items will be used to calculate pHDI and nHDI. The pHDI includes 10 food items: wholemeal bread/bread rolls, coarse-ground groats, milk, fermented milk drinks, cheese curd products, white meat, fish, legumes, fruit, and vegetables. The nHDI includes 14 food items: white bread and bakery products, white rice and fine-ground groats, fast foods, fried foods, butter, lard, cheese, cured meat/smoked sausages/hot dogs, red meat, sweets, tinned meats, sweetened carbonated or still drinks, energy drinks, and alcoholic beverages. Daily frequencies of the consumption of the selected 10 (pHDI) or 14 (nHDI) food items will be summed up and recalculated into a range of 0%–100% points according to the questionnaire’s manual [30]. A higher percentage of points reflected greater adherence to the diet quality score, i.e., for pHDI – better diet quality and for nHDI – worse diet quality.

### 2.6. Other Lifestyle Factors 

Self-reported data related to physical activity, time spent in front of a screen, and sleep duration (over the 12 last months), as well as smoking status will also be collected with the KomPAN questionnaire [30,31]. 

Physical activity in leisure time will be determined by choosing one of three categories: low (sitting, watching TV, reading, light housework, walking less than 2 h a week), moderate (walking, cycling, moderate exercise, working at home/garden, or other light physical activity performed 2–3 h/week) or high (cycling, running, gardening, and other sports/recreational activities requiring physical effort over 3 h/week). Physical activity at work will be determined by choosing one of three categories: low (more than 70% of time spent sedentary), moderate (about 50% of time spent sedentary and 50% active), or high (about 70% of time spent active or physical labor of high intensity). Screen time, defined as time spent watching TV or using computer/tablet (including work), will be determined by choosing one of six categories: <2 h, 2 to <4 h, 4 to <6 h, 6 to <8 h, 8 to <10 h, 10 h or more. Sleep duration during weekdays will be determined by choosing one of three categories: ≤6 h/day, >6 to <9 h/day, or ≥9 h/day. The same categories will be chosen for sleep duration during the weekend. Smoking status will be collected for the past and present (yes, no), and dichotomous categories will then be created: (i) non-smoker or past smoker, (ii) current smoker.

### 2.7. Quality of Life (QoL) 

To assess QoL, a thyroid-specific questionnaire (ThyPRO) in a Polish version (ThyPROpl) will be applied [32]. There is substantial evidence for the clinical validity and reliability of the questionnaire in patients with mild thyroid disorders [33,34]. The ThyPRO consists of 84 items summarized in 13 scales as well as a single item measuring the overall impact of thyroid disease on QoL within the last four weeks. The questionnaire covers physical symptoms specifically relevant to thyroid diseases (e.g., symptoms of hyperthyroidism and goitre) and non-specific aspects of high importance to patients with thyroid diseases (e.g., fatigue). Responses for each item will be scored with points on a five-point Likert scale as follows: ‘not at all’ (0 points), ‘a little’ (1 point), ‘some’ (2 points), ‘quite a bit’ (3 points), and ‘very much’ (4 points). The average score of items in a scale will be divided by four and multiplied by 100 to yield 13 scales ranging from 0 to 100 points [32,33], with higher point scores indicating a worse quality of life related to each scale.

### 2.8. Nutrition Knowledge and Sociodemographic Factors

Nutrition knowledge will be assessed with the KomPAN questionnaire [30,31]. Participants will be asked to provide answers (true/false/unsure) to the set of 25 statements; 1 point will be assigned for every correct answer and 0 points will be assigned for the wrong answer or ‘unsure’. All points will be summarized to express nutrition knowledge score (range 0–25 points). 

Data regarding sociodemographic variables will be obtained with the KomPAN questionnaire. Age (in years) will be recorded. The following variables will be collected using closed structured questions: place of residence (village; town <20,000 inhabitants; town 20,000 to 100,000 inhabitants; city over 100,000 inhabitants), level of education (primary; lower secondary; higher secondary; higher), financial situation (below average; average; above average), and household situation (we live very modestly; we live modestly; we live normally; we live relatively wealthy; we live very wealthy – all categories with detailed written explanation).

### 2.9. Adiposity Markers 

The measurements of body weight (kg; with electronic digital scale SECA 799), height (cm; with a portable stadiometer SECA 220), waist circumference (WC, cm; with a stretch-resistant tape SECA 201) will be taken. Body fat (BF, %) and skeletal muscle index (SMI, %) will be assessed using the bioelectrical impedance analysis (BIA) method (with the body composition analyzer SECA mBCA 525). All measurements will be taken in light clothing and without shoes according to the International Standards for Anthropometric Assessment (ISAK) guidelines [35]. Body mass index (BMI, kg/m^2^) and waist-to-height ratio (WHtR) will be calculated. Central adiposity (central obesity) will be identified as WHtR ≥0.5 [36] or WC ≥80 cm (ethnic- and gender-specific threshold) [37]. General adiposity will be identified with body mass index (overweight BMI = 25–29.9 kg/m^2^, obesity BMI ≥30 kg/m^2^) or body fat percentage (obesity gender-specific threshold: BF >33%) [38,39,40]. For a comprehensive interpretation of excessive body mass, which may result from excessive body fat mass and/or high skeletal muscle mass, low skeletal muscle index will be identified as SMI <22% (gender-specific threshold) [41]. 

### 2.10. Thyroid Function Tests (TFTs) and Metabolic Parameters

The following tests will be applied to monitor thyroid function: serum concentration of thyrotropin (TSH), free thyroxine (FT4), free triiodothyronine (FT3), serum auto-antibodies against thyroid peroxidase (TPOAb), and thyroglobulin (TgAb) [4,26]. TFTs results will be collected from the respondents (from different laboratories) and will be interpreted using laboratory thresholds to find abnormalities.

Office measurements (i.e., taken at a dietetic lab) of the systolic (SBP) and diastolic (DBP) blood pressure (mmHg; with an electronic shoulder manometer and an automatic analyzer of the ankle brachial index MESI ABPI MD by the oscillometric technique) will be taken [42,43]. Serum fasting glucose (FBG), triglycerides (TG), and total cholesterol (TC) will be measured in capillary blood (with an Accutrend PLUS analyzer, Roche Diagnostics). Metabolic parameters will be determined as elevated if FBG ≥100 mg/dL, TG ≥150 mg/dL, TC ≥200 mg/dL [44,45] and at least one component of blood pressure is elevated (SBP ≥130 or DBP ≥85 mmHg) [44]. 

### 2.11. Sample Size

The minimum sample size was calculated based on the expected improvement in overall QoL (decrease in points) and BMI (decrease in kg/m^2^) after the six-month follow-up and the expected difference between the experimental and the control group. A decrease in overall QoL and BMI with a 50% difference between groups (in the experimental/control group, a decrease by 20/10 points and 2.0/1.0 kg/m^2^, respectively) was assumed. For this calculation, a 5% significance level and 80% power were considered. The sample size required at baseline is approximately 100 respondents (for the experimental/control group: 50/50 for overall QoL and 42/42 for BMI), including a 20% drop-out rate at the end of the study. When the data collection is completed, the adequacy of the study sample will be checked by applying the post hoc statistics to calculate power.

### 2.12. Statistical Analysis

Categorical variables will be presented as a sample percentage (%) and continuous variables will be presented as means with a 95% confidence interval (95%CI) or medians with an interquartile range (IQR) for variables with normal or non-normal distribution, respectively. To assess the impact of dietetic counseling and verify the differences between the experimental and the control group, two-tailed tests will be applied; *p*-values < 0.05 will be considered significant. 

The normality of variable distribution will be verified with a Kolmogorov–Smirnov test before the statistical analysis. For continuous variables, e.g., diet quality scores, AdhS, scores of QoL, nutrition knowledge score, TFTs, markers of adiposity, metabolic parameters, and changes after a six-month follow-up with respect to baseline will be verified with a T-test for dependent samples or a Mann–Whitney test, for variables with normal or non-normal distribution, respectively. For categorical variables, logistic regression modeling will be applied to determine the chance of the occurrence of high diet quality, high AdhS, high QoL, as well as low adiposity and elevated metabolic parameters. The odds ratios (ORs) and 95% confidence intervals (CIs) will be calculated. The significance of ORs will be verified with Wald’s statistics. A crude model and models with an adjustment for confounders will be created. All analyses will be performed with Statistica software (version 13.3 PL; StatSoft Inc., Tulsa, OK, USA; StatSoft, Krakow, Poland).

## 3. Discussion

The recommendations in the ‘Diet4Hashi’ protocol are based on scientific evidence related to Hashimoto thyroiditis and hypothyroidism, as described in detail in Section 3.1 and Section 3.2. They were also inspired by guidelines developed for the general population, including Polish guidelines as described in the Pyramid of Healthy Nutrition and Physical Activity [27].

### 3.1. Basis for Development of a Self-Monitoring Diary for Recommended Foods (A)

#### 3.1.1. Vegetables 

The consumption of vegetables is recommended several times a day, which is in line with the recommendation for the general population [27,28,29]. Various kinds of vegetables are recommended, although attention should be paid to the consumption of soybean-related foods and raw cruciferous vegetables (see Section 3.2). The special function of vegetables for HT subjects can be attributed to phytosteroles. These compounds are present in vegetables in moderate amounts and have shown immunomodulatory and anti-inflammatory properties [46]. 

#### 3.1.2. Foods Rich in Calcium

For foods rich in calcium (e.g., milk, fermented milk drinks, curd cheese, cheese), the recommended consumption frequency is several times a day, which is in line with the recommendation for the general population [27,28,29]. Calcium deficiencies can have serious consequences for the health of women with hypothyroidism. The main role of thyroid hormones is metabolism regulation, but it also participates in bone tissue reconstruction. One consequence of hypothyroidism is a reduction in the processes of bone tissue reconstruction resulting from the inhibition of bone mineralization. Therefore, the diet of patients with hypothyroidism should be rich in dietary calcium, such as milk, yoghurt, cheese, low-, regular-, or high-fat dairy, and small fish eaten along with bones [47]. Some studies on HT have focused on lactose intolerance diagnosed with an elimination diet, hydrogen tests, lactose tolerance tests, or small intestinal biopsies [16]. It was found that 75% out of 84 patients with HT, all of whom were residents of the Mediterranean Sea region and Turkey, had lactose intolerance. Asik et al. [48] found that lactose elimination from the diet of HT patients decreased the TSH level without a modification dose of levothyroxine. For this reason, the authors suggested that patients demonstrating great fluctuations in the concentration of TSH while taking a steady dose of levothyroxine should be tested for lactose intolerance.

#### 3.1.3. Fruits

Fruit consumption is recommended at least once a day, similar to the recommendation for the general population [27,28,29]. Fruits and vegetables are an abundant dietary source of polyphenols and micronutrients in the human diet (if consumed). They are known for their anti-inflammatory, immunomodulatory, and antioxidant effects in the body [49,50]. Tonstad et al. [51] reported an association of a vegan diet (rich in fruits and vegetables) and a lower risk of hypothyroidism. In 2014, a meta-analysis of 19 case-control studies on dietary factors and thyroid cancer risk found a weak inverse association with vegetable consumption and no association with fruit consumption [52]. 

#### 3.1.4. Whole Grains

The recommended consumption of whole grains, e.g., buckwheat grain, wholemeal wheat, and rye bread, at least once a day, is in line with the recommendation for the general population [27,28,29]. A gluten-free diet has been given a lot of attention with respect to HT. The literature shows that celiac disease is associated with an increased prevalence of autoimmune thyroid disease (AITD) and vice versa, with a prevalence of up to 9% of adults with AITD [53,54]. In celiac disease patients, the prevalence of hypothyroidism is 2% to 5% [55]. Krysiak et al. [17] suggest that a gluten-free diet may bring clinical benefits to subjects with autoimmune thyroid disease. The study included 34 Polish women with recently diagnosed and previously untreated AITD. Roy et al. [56] in a pooled analysis, including 6024 patients with AITD, found a markedly increased prevalence of biopsy-confirmed coeliac disease. The authors concluded that all patients with AITD should be screened for the presence of coeliac disease. Sategna et al. [57] observed that a gluten-free diet can reverse the abnormality of subclinical hypothyroidism, although they did not find a correlation between the time of exposure to gluten in celiac disease individuals and the risk of autoimmune diseases. Other authors also reported that a gluten-free diet has a protective effect against thyroid diseases, and they recommended that the gluten-free diet should begin early, before autoimmune disorders are established, in order to prevent or minimize their development [58]. To the contrary, Ch’ng et al. [53] disagreed with these findings, arguing that there is little evidence to support the use of a gluten-free diet to reduce the development of AITD. Following this data, a gluten-free diet is still controversial as to its effectiveness in preventing AITD and hypothyroidism. However, for the treatment of celiac disease, it is mandatory to follow a gluten-free diet [19,20,21]. Further investigation is necessary for a better understanding of the role of gluten in thyroid diseases. Currently, the recommendation for whole grains results from the content minerals, vitamins, and dietary fiber in these foods [27,28,29]. The consumption of whole grains can also improve selenium intake, e.g., whole-wheat flour contains 13.6 μg of selenium per 100 g (24.7% of recommended dietary allowance (RDA)) [27]. The average content of selenium in foods based on popular gluten-free cereals was 2.8 μg/100 g, and in foods based on oat, amaranth, teff, and quinoa, it was 10.8 μg/100 g [59].

#### 3.1.5. Animal Foods Rich in Zinc

The consumption of animal foods rich in zinc (e.g., meats and eggs) several times a week is recommended. This is in line with the recommendation for the general Polish population, who usually consume too much red and processed meats [60,61]. The RDA of zinc is 8 mg/day for Polish women, and this amount can be found in meats in amounts varying from 1.2 to 8.4 mg/100 g [28]. There are reports that zinc deficiency is one of the causes of subclinical hypothyroidism [62]. A previous study by Ertek et al. [63] suggested that zinc deficiency leads to a reduction in the level of FT4 and FT3 and the development of hypothyroidism symptoms. In humans, zinc supplementation brought the thyroid function back to normality in hypothyroidean patients. The majority of studies on zinc and the HT were focused on zinc supplementation, but these findings were not considered due to being out of the study aim [64,65,66]. 

#### 3.1.6. Animal Foods Rich in Selenium

The authors’ recommendation to consume animal and plant foods rich in selenium is in line with previous findings that diets rich in selenium may increase the production of active thyroid hormone and reduce TgAb and TPOAb [67,68,69]. Stoffaneller et al. [70] showed that selenium intake and selenium status was suboptimal in European and Middle Eastern countries, with lower levels in the Middle East. The highest selenium status of all studies in the European region was found in Poland, where boys had a mean serum selenium concentration of 111.1 μg/L (optimal level: 289.9 μg/L) [71,72]. Lower serum selenium levels were found in the patients with AITD (60–99 μg/L) [70]. The most recent meta-analysis found that the use of selenium supplements reduced serum TPOAb levels after 3, 6, and 12 months in HT patients treated with levothyroxine and after three months in untreated cases [73].

The authors have distinguished animal and plant dietary sources of selenium for three reasons: (i) to simplify recommendations for the patient, (ii) to reduce the chance of any unilateral exchange between both food groups and maintain a large variety of diet, and (iii) plant sources of selenium, i.e., nuts and seeds, are also rich in zinc.

Fish and seafood are indicated as animal foods rich in selenium with a recommended consumption of several times a week. This is in line with the recommendation for the general population, although a guideline on eating fish and seafood at least twice a week (including fatty fish at least once a week) is mainly based on the content of *n-3* polyunsaturated fatty acids in these foods [27,28,29]. Seafood and organic meats are good sources of selenium, followed by muscle meats [68,69]. 

#### 3.1.7. Nuts and Seeds

Various kinds of nuts and seeds are recommended to be consumed several times a week. These foods are a good source of selenium and zinc (both important in thyroid metabolism) and also rich in dietary fiber. Brazil nuts are the richest food source of selenium, although they cannot be recommended as a main selenium source because they are generally not a commonly eaten food, and selenium content is highly variable, ranging from 0.03 to 512 mg/kg fresh weight [68,74,75]. In general, in all kinds of nuts, the zinc concentration ranges from 2.1 to 4.7 mg/100 g [27,28,29]. Farhangi et al. [76] showed the potential beneficial effect of powdered seeds of *Nigella sativa* in improving thyroid status and anthropometric variables in patients with HT. Nevertheless, its function on the human thyroid requires more trials, and this finding should be interpreted with caution. The thyroid hormone is made by combining iodine and tyrosine. In a balanced diet, a person should receive enough of this amino acid, but tyrosine intake can be supported by the consumption of pumpkin seeds, sesame seeds, and almonds [27,28,29].

### 3.2. Basis for the Development of a Self-Monitoring Diary for Foods with Limited Consumption (B)

#### 3.2.1. Raw Cruciferous Vegetables

The consumption of raw cruciferous vegetables, e.g., kale, bok choy, white cabbage, red cabbage, broccoli, brussels sprouts, and cauliflower should be limited (once a week or less). Cruciferous vegetables, as well as soybean-related foods, contain goitrogens, which interfere with thyroid hormone production and utilization. A goiter may be a response to an overactive or underactive thyroid gland. Unless there is a co-existing iodine deficiency, these foods are generally of no clinical significance according to the statement of the Institute of Medicine [77]. Goitrogens are inactivated by heating and cooking—the thermal processing leads to inactivation of about 30% of goitrogens. [77]. Thus, it is important to educate patients on how to prepare these vegetables. Matana et al. [23] showed that a dietary group with a higher consumption of root vegetables, flower vegetables, leafy vegetables, fruity vegetables, and legumes was negatively associated with plasma TP-Ab and/or TgAb (upper tertile OR = 0.88, 95% CI 0.78–0.99 versus bottom tertile, *p* = 0.048).

#### 3.2.2. Sweets, Sugar, and Honey

Limiting the consumption of sweets, sugar, and honey including high-sweetened jam and fruits candied (once a week or less) is recommended, as these foods are a source of monosaccharides and many of these foods contain a large amount of saturated fatty acids and trans-isomers of unsaturated fatty acids. This recommendation is in line with the recommendations for the general population (no precise cut-off for Poles was given) to prevent obesity, diabetes, and other diseases, including cancers [27,28,29]. It should be underlined that the risk of the development of diabetes is greater in HT patients than people without thyroid diseases. Although the autoimmune character of the HT predisposes to type 1 diabetes, pro-inflammatory cytokines also have an influence on receptor insulins, which can lead to impairment of their function and the development of insulin resistance and type 2 diabetes [13]. 

#### 3.2.3. Sweetened Beverages and Energy Drinks

Limiting the consumption of sweetened beverages energy drinks (once a week or less) regardless of the sweeteners type (sugar, glucose–fructose syrup or artificial sweeteners, etc.) is recommended. Sweetened beverages can affect hormonal levels in the blood by increasing T4 and parathyroid hormones, and lowering T3 and aldosterone levels [78]. Abbott et al. [22] recommended the elimination of foods with low nutritional value (sugar-sweetened beverages, ultraprocessed foods, etc.), and foods that could result in an aberrant immune response via dysregulated antigen presentation or detrimentally affect both the gut microbiome and the integrity of the gastrointestinal barrier. Animal studies report that artificial sweeteners affect the immune system and show that sucralose diminishes the thyroid axis activity [79]. Energy drinks, regardless of whether they are standard or low-calorie drinks, commonly contain sugar substitutes such as saccharin, aspartame, sucralose, acesulphame K, and neotame. Clinical reports show that the intake of artificial sweeteners may play a role in AITD and lead to increased TSH [79].

#### 3.2.4. Fast Food

Limiting the consumption of fast food (once a week or less) is recommended. The recommendation for the general population is to avoid fast food [27,28,29]. There is a growing, alarming trend of fast food consumption associated with a worsening of cardiometabolic outcomes, including obesity [80]. It was reported that the consumption of fast foods ≥2 times/week increased the risk of insulin resistance [81]. In adults, every one-meal/week increase in fast food and sit-down restaurant consumption was associated with an increase in BMI [82].

#### 3.2.5. Soybean and Millet

Soybean and millet—e.g., soya: seeds, sprouts, tofu, milk; millet groats—are controversial foods. Limiting the consumption of these foods (twice a month or less) is recommended, despite the potential human health benefits of soy in the prevention of cancer, cardiovascular diseases, the reduction of menopause symptoms, increased bone-mineral density, and decreased insulin resistance [83]. Conversely, soy has raised concern about thyroid gland function [84]. According to Messina et al., the literature provides little evidence that the consumption of soy foods or isoflavones has adverse effects for euthyroid individuals with iodine deficiency. However, there is a theoretical concern based on in vitro and animal studies that individuals with impaired thyroid function and/or whose iodine intake is marginal are more likely to develop hypothyroidism with soy consumption [85]. Thus, it is vital that consumers of soy-based foods be sure that their iodine intake is adequate. In addition, soy may hinder the absorption of thyroid drugs [84].

#### 3.2.6. Alcohols

Limiting the consumption of alcohol (once a month or less) is recommended. However, the recommendation for the general population is not to consume alcohol [27,28,29]. Moderate alcohol consumption has been shown to have a protective role in the development of the HT [86,87], while regular alcohol use did not change TPO-Ab, but may risk HT hypothyroidism [86]. A population-based case-control study in Denmark likewise observed that moderate alcohol consumption reduced the risk of overt autoimmune hypothyroidism: ORs were 1.98 (95% CI, 1.21 to 3.33) for 0 alcoholic units/week, 1.00 for 1 to 10 units/week (reference), 0.41 (95% CI, 0.20 to 0.83) for 11 to 20 units/week, and 0.90 (95% CI, 0.41 to 2.00) for ≥21 units/week [88].

### 3.3. Strengths and Limitations

The strength of the study was the development of a simple dietary protocol with a user-friendly approach that is easy to use for both individuals with HT and health professionals. This dietary protocol was developed based on scientific evidence and presents the best adherence to current knowledge. Simple markers of adiposity (BMI, WHtR) and metabolic parameters (FBG, TG, TC) useful in dietetic counseling and clinical practice (and easy to replicate in other studies) will be applied. In addition, advanced methods of adiposity measurement (bioelectrical impedance analysis method with an SECA mBCA 525 analyzer) will be applied to accurately measure body composition, including fat mass and skeletal muscle mass. Since the data are experimental observations, the causal effect of dietary intervention can be demonstrated. 

The main limitation is that the study will not have a controlled placebo as there is no ‘dietetic placebo’ for dietetic counseling. Therefore, the graphic–text tool will be controlled with conventional dietetic management. Secondly, the study is not double-blinded because (i) the researcher who will be involved in dietetic counseling will be aware of which participants are included in the QDP group and which participants are in the CDC group, and (ii) it is possible that respondents may be aware of what dietetic counseling they are following. The next limitation is the use of non-thyroid-specific food frequency questionnaires (FFQs) to describe diet quality. Both FFQs have been previously used to study the association between diet and health outcomes in the context of non-alcoholic fatty liver disease [89], celiac disease [90], metabolic syndrome [91], breast and lung cancers [92,93], and are accurate tools to describe overall diet quality as well as negative (nHDI) or positive (pHDI) aspects of diet quality. Unfortunately, both FFQs do not contain any detailed specification of food items that are important (more or less recommended for consumption) in the ‘Hashi diet’ (e.g., buckwheat grain or cruciferous vegetables). This limitation will be reduced by a quantitative (in points) assessment of the consumption frequency of those foods, which was recorded in real time via a self-monitoring diary, to assess adherence to the QDP. Further limitations can be attributed to using FFQs. Although there is evidence of many advantages related to data collection with the FFQs, several shortcomings have been reported, e.g., the data relies heavily on memory; therefore, declining cognitive ability may result in errors when reporting on food frequency consumption; the food list cannot cover all the foods consumed by respondents, which may lead to underreporting; and the use of a questionnaire causes some uncertainty due to social desirability bias, especially in females, who may provide more socially acceptable answers [94]. To describe sedentary (screen time) and active behaviors (at leisure time and work), a questionnaire with simple questions will be used. This will allow respondents to be ranked into categories of low and high physical activity levels, but will not allow precise measurement of physical activity. Office measurements of blood pressure will be taken, so ‘white coat’ hypertension phenomena should be taken into account when elevated blood pressure is interpreted [42,43]. Future studies should consider applying other methods to fully describe subjects’ dietary behaviors (e.g., biomarkers and/or repeated food records) and measure physical activity (e.g., accelerometry) instead of self-reported data [94,95] and also implement repeated blood pressure measuring (e.g., 24-hour monitoring) [42,43]. 

## 4. Conclusions

Due to the development of an easy-to-follow qualitative dietary protocol for HT subjects (Diet4Hashi), this study will contribute to providing valuable data for dieticians and physicians. It is anticipated that this graphic-text qualitative dietary protocol, by improving food selection and diet quality, can reduce adiposity and improve metabolic parameters and the quality of life of HT women. Further investigations are needed to verify the use of this dietary protocol in HT males. 

## Figures and Tables

**Figure 1 ijerph-16-04841-f001:**
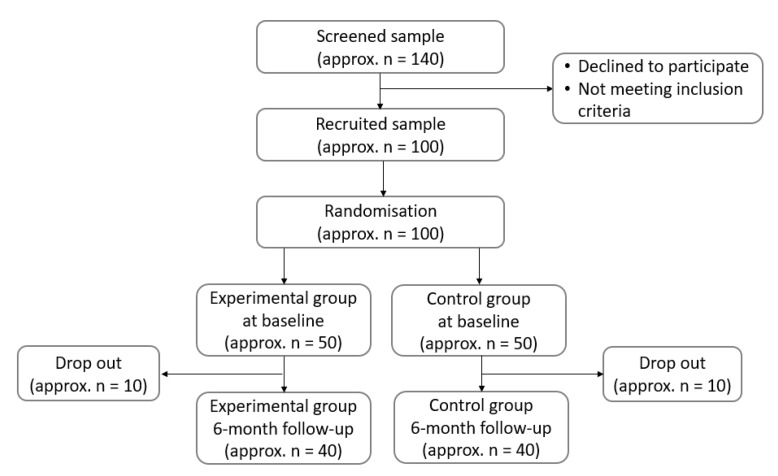
Sample collection chart.

**Table 1 ijerph-16-04841-t001:** Timetable of data collection.

Timing	Activities	Group	
		Experimental	Control
T0 visit (enrolment)	Eligibility screening; Written informed consent; Assignment to the experimental or control group.	✕	
T1 visit (baseline)	Data collection of diet quality and other lifestyle factors, quality of life, nutrition knowledge, sociodemographic factors.	✕	✕
	Measurements of adiposity, metabolic parameters, and thyroid function.	✕	✕
	Dietetic counseling.	QDP	CDC
Between visits	Home collection of dietary data with self-monitoring.	✕	
T2 visit (1-month follow-up)	Dietetic consultation to boost compliance with dietary guidelines.	✕	✕
	Data collection of diet quality and other lifestyle factors, quality of life.	✕	✕
	Measurements of adiposity and metabolic parameters.	✕	✕
Between visits	Home collection of dietary data with self monitoring.	✕	
T3 visit (6-month follow-up)	Data collection of diet quality and other lifestyle factors, quality of life, nutrition knowledge.	✕	✕
	Measurements of adiposity, metabolic parameters, and thyroid function.	✕	✕

QDP—a qualitative dietary protocol; CDC—conventional dietetic counseling.

**Table 2 ijerph-16-04841-t002:** Participant eligibility criteria. BMI: body mass index.

**Inclusion Criteria**: women age >18 and <60 years; prior diagnosed Hashimoto thyroiditis: hypothyroidism or subclinical hypothyroidism; medical certificate for levothyroxine treatment or other thyroid hormones; agreement to take part in the study;
**Exclusion Criteria**: non-Hashimoto thyroiditis; pregnant or breastfeeding women; current/past use of dietary supplements, e.g., selenium or iodine; current use of other supplements in an amount above an upper level; diagnosed presence of other comorbidities: other endocrinopathy and metabolic diseases (e.g., diabetes, cardiovascular diseases), other thyroid diseases with chronic inflammation, diseases requiring an elimination diet (e.g., celiac disease, non-celiac gluten sensitivity, food allergy, food intolerance, etc.); other special diet with or without disease diagnosis lasting >4 months within the last 12 months: rigorous elimination diet (e.g., gluten-free, lactose-free), unbalanced diet (e.g., high protein, ketogenic), low-energy diet with an energy restriction >1000 kcal/day, very low-energy diet (e.g., liquid meal replacements), so-called ‘the autoimmune protocol’ and others that significantly change metabolism; underweight (BMI <18.5 kg/m^2^) or obesity with BMI ≥35 kg/m^2^; weight loss >4 kg in the last month.

**Table 3 ijerph-16-04841-t003:** The qualitative dietary protocol: (A) Self-monitoring diary for recommended foods.

Foods Recommended for Consumption Control the Frequency of the Food Consumption Every Meal/Eating Episode (or at least Every Day) and Check Adherence to Recommendations. Minimum Frequencies Are Indicated.							
Day	Vegetables (see diary B)	Foods rich in calcium e.g., milk, fermented milk drinks, curd cheese, cheese	Fruit	Whole grains e.g., buckwheat grain, wholemeal wheat and rye bread	Animal foods rich in zinc e.g., meats: beef, lamb, pork; eggs	Animal foods rich in selenium e.g., fish, seafood	Nuts and seeds e.g., nuts, pumpkin seeds, sunflower seeds
	Several times a day	Several times a day	Once a day	Once a day	Several times a week	Several times a week	Several times a week
Month: **Mark** ✕ **each time after food consumption**							
1							
2							
3							
…							
31							
*Month: June 2019 **Sample record of the Patient***							
*1*	✕✕	✕	✕✕	✕		✕	
*2*	✕✕✕	✕	✕	✕	✕	✕	
*3*	✕	✕✕✕	✕	✕		✕	
*4*	✕	✕✕		✕	✕	✕	
*5*	✕✕✕	✕	✕	✕	✕	✕	
*6*	✕✕	✕✕				✕	
*7*	✕✕✕	✕✕	✕		✕		

**Table 4 ijerph-16-04841-t004:** The qualitative dietary protocol: (B) Self-monitoring diary for foods with limited consumption.

Foods with Limited Consumption Control the Frequency of the Food Consumption Every Meal/Eating Episode (or at least Every Day) and Check Adherence to Recommendations.Maximum Frequencies Are Indicated.
Day	Raw cruciferous vegetables e.g., kale, bok-choy, white cabbage, red cabbage, broccoli, brussels sprouts, cauliflower	Sweets, sugar and honey including high-sweetened jam, fruits candied, etc.	Sweetened beverages and energy drinkse.g., with sugar, glucose-fructose syrup, artificially sweetened	Fast foods e.g., hot-dogs, hamburgers, chips, French fries, pizza, tortilla, deep-fried foods	Soybean and millet e.g., soya: seeds, sprouts, tofu, milk; millet groats	Alcoholse.g., wine, beer, alcohol drinks, vodka, brandy
Once a week	Once a week	Once a week	Once a week	Twice a month	Once a month
Month: **Mark** ✕ **each time after food consumption**
1						
2						
3						
…						
31						
*Month: June 2019 **Sample record of a Patient***
*1*	✕					
*2*			✕			
*3*	✕					
*4*						
*5*		✕				
*6*						
*7*			✕	✕	✕

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
