# Peer review of "Evaluation of Qualitative Dietary Protocol (Diet4Hashi) Application in Dietary Counseling in Hashimoto Thyroiditis: Study Protocol of a Randomized Controlled Trial"

_ijerph, 2019, doi:10.3390/ijerph16234841_

Round 1
Reviewer 1 Report
Overall, this manuscript describes a very thorough and well designed study and will add significantly to the literature around dietary counseling and dietary approaches to HT.
I would suggest some slight revisions to the abstract that would then carry into the paper itself. First a simple suggestion, I would not say metabolic abnormality; what it seems you are really referring to are simply metabolic or cardiometabolic parameters/indices and I would suggest using one of those terms rather than abnormality unless you are specifically describing the metabolic parameter or lab in an abnormal state. I would also suggest some revisions with the main described aims of the study, to ensure people reading understand the study design in a clear concise manner. It was a little challenging to discern the exact difference between the two study arms in some sections of the paper including the abstract, and the paper would benefit from more clearly outlining the difference in the two interventions. I was under the impression both groups are going to be told to eat the same things, but one group is going to be given a different graphical representation and there would be no other elements of the counseling or duration that would differ? I would seek to make this quite clear and delineated early in the abstract and have consistency throughout the paper.
The introduction was overall quite robust and I only have a few small suggestions. First I was a little confused about this statement "Women had thyroid gland 47 disease almost 9 times more often than men, with an increasing tendency" and what the "an increasing tendency" was included for and represented? and would otherwise remove this. Second, I think if you are going to begin introducing certain studies and dietary approaches to HT as you describe regarding gluten, Vitamin D, iodine, etc, the article would be strengthened by including other interventional studies including individuals with HT in the introduction and saving associative, epidemiological and correlative studies for later in the discussion as you did. For example, the article's introduction may be strengthened by including a further description of the article by Abbott et. al. in April 2019 that was an interventional study in individuals with HT looking specifically at quality of life and thyroid hormones, Ab's as you are in your study. Otherwise the introduction was strong and added value to the manuscript as a whole.
I found the Methods Section to be quite sound in describing in detail the study design, the matriculation process, the elements of data collection and statistical methods employed. I was curious to understand more the rationale for selecting the specific dietary scores as it was unclear what benefit there would be in calculating a mediterranean diet score from the respective food frequency questionnaires. I would suggest focusing any comparative dietary scores on the focus elements of the diet4hashi graphic as you later describe with the foods to promote and the foods to limit, and specifically evaluate changes and patterns of consumption of these foods as provided in the education rather than a larger dietary pattern that is not as directly related to the dietary suggestions you are providing. It's not as helpful to calculate a score for instance, incorporating certain elements of legume consumption that is not incorporated in the diet4hashi recommendation and I would argue that the correct promotion of animals foods rich in zinc, which would include red meat, would led to a lower/less adherent Mediterranean diet score and skew data here.
I also wanted to clarify the study's inclusion and exclusion criteria as it appeared that you were requiring thyroid ultrasounds and also that people are on replacement medication? Would individuals with elevated antibodies, but not on thyroid replacement be allowed in the study? This criteria seemed a little bit strict and perhaps it would prove challenging to obtain 100 individuals meeting those specific criteria- all with previous US information. I would also likely modify the criteria talking about diseases of chronic inflammation as HT itself is a disease of chronic inflammation and I would only focus on excluding individuals with organ failure related to some other inflammatory process.
In the discussion section, there will clearly continue to be debate about whether gluten containing whole grain consumption should be promoted as well as dairy products as a source of calcium. The criteria for Diet4Hashi appear set and you have done a decent job articulating the evidence for the inclusion or exclusion of gluten containing grains, but it does appear at least from the evidence you cite and focus on that the intervention would likely be better served by focusing only on non gluten containing grains if whole grains are to be promoted. The promotion of certain dairy products is also something to potentially be challenged and is controversial for inclusion in the recommendations. I agree with the strengths and limitations outlined in the discussion and believe that provides a fair assessment of the study's intentions and capacities.
With some modifications to the abstract and manuscript as described, I believe the paper will be strengthened and provide valuable data and insight to the scientific community.
Author Response
|
* Corresponding author: Natalia Wojtas (Ulewicz), MSc University of Warmia and Mazury Department of Human Nutrition Sloneczna 45f, 10-718 Olsztyn, Poland Tel. +48 (89) 524 5514 E-mail: natalia.wojtas@uwm.edu.pl |
Authors’ Response to the Reviewers’ Comments
Journal: International Journal of Environmental Research and Public Health
Manuscript: ijerph-640278
Title: Evaluation of qualitative dietary protocol (Diet4Hashi) application in dietary counselling in Hashimoto thyroiditis: study protocol of a randomised controlled trial
Authors: Natalia Wojtas (Ulewicz), Lidia Wadolowska, Elżbieta Bandurska-Stankiewicz
Article type: protocol
27th November 2019
Dear Ms. Layla Wang, Dear Reviewers,
We are very grateful that we have the opportunity to present our manuscript (ID: ijerph-640278) entitled ‘Evaluation of qualitative dietary protocol (Diet4Hashi) application in dietary counselling in Hashimoto thyroiditis: study protocol of a randomised controlled trial’.
We greatly appreciate the time and efforts taken by the Reviewers and the Editor to review our manuscript. We have addressed all issues indicated in the review reports, and believe that the revised version can meet the journal publication requirements.
Please find our responses to the Reviewer’s comments attached. The manuscript has been corrected for language errors, using professional editing (native speaker) and proof-reading service. All changes in the manuscript are highlighted in blue font.
Yours Sincerely,
Natalia Wojtas (Ulewicz)

Reviewer 2 Report
This is a very well constructed protocol for an RCT. Authors tried to develop a dietary protocol (Diet4Hashi), which is easy to follow for HT subjects that will contribute to providing valuable data, which is useful to dieticians and physicians.
There are some points to be revisited before this paper is published.
Authors should indicate more practical issues on how they developed the study protocol, where they were based on etc.
Sample size power calculation is needed to be more specific, based on current software. Authors claim that they were based on an improving overall QoL with a 50% difference between groups, but another more objective parameter might be more accurate – to base the power.
In pages 136 and over, more specific points have to be given that differentiate the 2 protocols and the rationale on which authors have been based to construct the new one. Also the disadvantages of the CDC should be mentioned that led to the construction of a new protocol.
The items after the discussion might be better if they were put in the materials section.
Author Response

(The authors gave the same response as above.)
